# Interaction Processes between Health Professionals and Moroccan Immigrant Women in Reproductive Healthcare: The Disagreement in the Encounter—A Qualitative Study

**DOI:** 10.3390/healthcare12161577

**Published:** 2024-08-08

**Authors:** María Idoia Ugarte-Gurrutxaga, Sara María Ulla Diez, Brígida Molina-Gallego, María Humanes-García, Gonzalo Melgar de Corral, Fernando Jesús Plaza del Pino

**Affiliations:** 1Faculty of Physiotherapy and Nursing of Toledo, Campus Toledo, Castilla-La Mancha University, 45003 Toledo, Spain; maria.ugarte@uclm.es (M.I.U.-G.); maria.humanes@uclm.es (M.H.-G.); gonzalo.melgar@uclm.es (G.M.d.C.); 2Ministry of Social Rights, Consumer Affairs and Agenda 2030, 28006 Madrid, Spain; saraulla@imserso.es; 3Faculty of Health Sciences, University of Almeria, 04120 Almeria, Spain; ferplaza@ual.es

**Keywords:** communication, cultural competence, immigrants, intercultural mediation, qualitative research

## Abstract

Introduction: Spain is a multicultural society and has been defined by several authors as an immigrant-receiving country. Moroccan women of childbearing age constitute 28.20% of Moroccan immigrants. Objectives: describe the interaction processes that occur between health professionals and Moroccan immigrant women in reproductive healthcare. Methods: Qualitative descriptive study based on Grounded Theory. Thirty immigrant women from Morocco and thirty-five health professionals participated in the study. Specific dimensions of analysis were defined and used to design the interview guide and focus groups. Results: In the healthcare encounter, the construction of an effective communicative space between the people involved in it is essential; however, the language barrier and the interpersonal relationships characterized by silence in the encounter make a meaningful healthcare relationship difficult for those who participate in it. Conclusions: There are communication and relationship problems that alter healthcare and the professional–patient relationship which require the use of translation programs, the incorporation of intercultural meters, and the development of cultural competence in health professionals.

## 1. Introduction

The present research addresses the topic of the reproductive health of Moroccan immigrant women, with the aim of describing and analyzing the experience and discourse that health professionals and Moroccan women have in the healthcare encounter in the perinatal stage.

Currently, societies are becoming increasingly multicultural. Spain is not immune to these circumstances and has been defined by various authors as a country that receives immigrants; in fact, according to the Report “Immigration in Spain: Effects and opportunities” [1], Spain is in fourth place in the European Union in terms of the volume migrants they receive.

According to data from the National Institute of Statistics [2], in 2022, the number of Moroccan immigrant women of childbearing age (15–49 years) was 289,354. The Immigration Report in Spain developed by the Economic Council, mentioned above, reflects that one of the most frequent reasons why the immigrant population goes to hospitals is related to pregnancy and birth care.

We have found studies that delve into the perception of Moroccan immigrants and discriminatory treatment towards them in the National Health System [3], which they consider to be the result of their skin color, difficulty with the language, or differences in health beliefs. But whatever the cause, the reality is that in the care of the immigrant population, conflicts—perhaps not a problem at first—are real. Specifically in relation to racism, “*it requires the amalgamation of actors, interests, positions, behaviors, attitudes, contradictions and possibilities, among other aspects, that occur in every conflict*” [4].

Another important point of view is that of health professionals. In the research by Pérez-Urdiales and Goicolea [5], professionals highlight the barriers and difficulties that immigrant women encounter in accessing health services in the Basque Country, among which the following stand out: the characteristics of women, the attitudes of the health professional, the functioning of the health system itself, and health policies.

However, we are missing studies that focus on the interpersonal relationships and communication processes between Moroccan women and health professionals.

There is plenty of scientific evidence on the impact of communication on the results of the healthcare encounter [6,7], with inequality in communication related to health disparities. One of the profiles most susceptible to suffering inequality in communication is that of the immigrant population [8], since, in contexts of cultural diversity, there are differences in communication between patients and health professionals. These differences can affect mutual understanding, diagnosis, and satisfaction with a person’s care. Some key elements include empathy and intercultural experience and the ability to communicate in the patient’s language, since linguistic asymmetry can limit understanding and generate misunderstandings [9]. To be culturally competent, healthcare professionals must implement various interpersonal and organizational strategies to overcome communication and understanding barriers that arise from racial, ethnic, cultural, and linguistic differences [10]. The provision of culturally competent care is especially necessary in the perinatal stage, since the process of pregnancy, childbirth, and the postpartum period is a risk factor for the health of the mother and the child [11].

In this study, we incorporate a “culturalist” approach to communication in the healthcare field [7], understood as a process of exchange, participation, and shared meanings, leaving behind the biomedical model and placing ourselves in a critical paradigm.

From this approach, the patient, holder of rights, takes center stage, assuming an active and participatory role in their care process. We talk about communication as a process that facilitates the encounter and intercultural dialogue between patient and professional.

Taking into account that our objects of study are Moroccan immigrant women in the perinatal stage, this communicative process is expected to be complex, since from the outset their linguistic competence is poor and, in addition, we find a high percentage of illiteracy among immigrant women, as well as cultural differences in relation to reproductive health. The vulnerability of migrant women in their reproductive stage is a consequence of the intersectionality of the social factors of the migratory context, among which the gender category has an important weight and places them in a precarious situation [12].

We have found studies that show an increase in barriers to access good reproductive health in women with few resources and in a situation of marginalization [13], a profile that matches the majority of migrant women (discrimination, social inequality, poverty, access to work precarious, family responsibilities…).

With these premises, a series of measures are necessary to resolve the different communication conflicts, misunderstandings, and relationships that may arise between women and health professionals in order to achieve a dignified approach and guarantee their right to health [14]. Precisely, this is the objective of intercultural mediation in the healthcare field [15].

Most of the research to which we have had access has a quantitative methodological approach. It provides data on morbidity, care, or other dimensions [13] but does not address the communication processes between health professionals and the women who are the subject of our study.

Therefore, our objective is to detail the interaction processes that occur between health professionals and Moroccan immigrant women in reproductive healthcare.

## 2. Materials and Methods

### 2.1. Study Design

This is a qualitative descriptive study based on Grounded Theory. Grounded Theory allows us to build theoretical concepts as we explore the experiences and perspectives of our participants [16]. We seek to deeply understand the meanings and patterns that emerge from the data collected. The aim is to understand underlying discourses, opinions, and ideas, following the analysis steps described by Graneheim and Lundman [17].

### 2.2. Sample and Configuration

Our sample was 30 immigrant women from Morocco (Table 1) and 35 health professionals (7 men and 28 women) from the fields of Primary Care and Specialized Care, with the following profiles: nurses, midwives, family medicine, pediatrics and gynecology-obstetrics (Table 2).

The inclusion criteria established for the women were being of Moroccan origin and a resident in Castilla-La Mancha (Spain); aged between 21 and 50 years; having been a resident in Spain for more than 1 year; with primary education or higher; and having had at least one child in Spain. The only exclusion criterion was being health professionals.

Linguistic competence in Spanish was initially considered as an inclusion criterion when configuring the study sample. However, once the phase of contact with the Moroccan women had begun, we realized that this would make it difficult to form a large sample, so we decided not to maintain it. We were assisted in translating the speech by two women. One of them took part in two of the interviews and the other in the two focus groups, so that we were able to obtain the details of the speech of those who did not speak Spanish.

The inclusion criteria established for the professionals were belonging to one of the following professional profiles: Primary Care Nursing, Maternity Nursing, Pediatric Nursing, Midwifery, Gynecology/Obstetrics, and Pediatrics; having professional experience of more than three years; not holding management or trade union positions; and working in the Castilla-La Mancha Health Service (SESCAM) in Primary Care/Hospital Care.

The selection of participants, both migrant women and professionals, was of ‘convenience’ using the ‘snowball’ method of recruitment from informal and formal networks of the lead researcher. This first contact was made by telephone.

For the women, the interviews were carried out in their homes and at the Moroccan Women’s Association, and in the case of the professionals, they were conducted in their workplaces: Health Centers and hospital consultations.

The interviews and focus groups were conducted by the principal researcher, a nurse and anthropologist with extensive experience in this type of technique. The interviewer introduced herself and provided information about the study and its purposes.

### 2.3. Data Collection

The data were generated through semi-structured in-depth interviews—40 individual and 5 group interviews of the two profiles in our study—and 2 focus groups, both with Moroccan women. They were carried out between June 2020 and May 2021. Table 3 presents an outline of the contents addressed in the interviews and focus groups.

The breadth and diversity of the sample profiles and the different techniques used ensured, in our opinion, the triangulation of sources and methods. The first three interviews were used to set up the final interview and focus group guides.

Before starting each interview and each focus group, the study, its objectives, possible applications, the importance of their (voluntary) participation, and the confidentiality of the treatment of the data collected were explained to them and they were informed that it was possible to withdraw from the study at any time. They were also asked if they were interested in having information about the results.

The interviews and focus groups were audio-recorded with the consent of the participants and then transcribed in full for analysis.

The interviewer introduced herself and provided information about the study and its purposes. The average duration of the interviews was 60 min.

In cases of greater language difficulty for the Moroccan women, the help of a translator was available.

We subjected the number of interviews and focus groups to the “information saturation criterion”. Theoretical saturation was reached when the information collected did not contribute anything new to the development of the properties and dimensions of the analysis categories. The criteria to determine saturation were as follows: (a) the integration and density of the theory (it was saturated when the greatest number of variations within the theory had been analyzed and explained and when the relationship between the emerging categories obeyed a logical scheme—explanation of the research problem); (b) the combination of the empirical limits of the data (saturation was reached when the researcher did not have access to other data that contributed to the development of the research); and (c) the theoretical sensitivity of the analyzer (the researchers’ ability to theoretically address the data) [16].

### 2.4. Data Analysis

Data processing was initially carried out by verbatim transcription of the recordings of the focus groups and interviews and the recording of non-verbal information collected by the observers. These transcripts and the extracts used were anonymized by assigning them the codes shown in Table 4. The recordings and transcripts were kept private and in the custody of the principal researcher.

Part of the research team participated in the analysis, namely, M.I.U.-G., M.H.-G., G.M.d.C., and F.J.P.d.P.

The Atlas-Ti 8 Programme (Scientific Software Development GmbH, Berlin, Germany) was used for the coding of all the information. Subsequently, the main dimensions around which the discourse is articulated were identified and then the most relevant aspects of each of the themes were differentiated, grouping them into categories that are shown in the Results Section.

The coding and categorization were checked by all members of the research team (M.I.U.-G.,B.M.-G., M.H.-G., G.M.d.C., S.M.U.-D., and F.J.P.d.P.) to reach a consensus. An external reviewer with expertise in conventional content analysis was brought in to verify the coding, interpretation, and data categorization process. 

Data collection and analysis was conducted between June 2020 and December 2023 to ensure sufficient engagement with the data [17].

### 2.5. Ethical Considerations

We have the Favorable Resolution of the Clinical Research Ethics Committee of the Integrated Health Area of Talavera de la Reina (Toledo, Spain) for the current study (CElm Code: 7/2019, dated 3 December 2019). The research did not entail risks for the participants, since the techniques used and their contents do not entail risks for their physical or psychological integrity.

All information collected was analyzed confidentially, guaranteeing the anonymity of the participants.

## 3. Results

The results are presented, articulated according to the two themes and the emerging categories of the speeches of the people participating in the study (Table 5).

As we can see below, two categories emerge in the discourse of the participants. On the one hand, the difficulties in communication due to language differences are highlighted and the existence and use/non-use of tools to overcome language barriers is addressed, and on the other hand, ‘the silence of women’ in the care encounter emerges as a category. This silence is interpreted differently by each of the profiles in the sample. It is of vital importance to take into account the different perceptions as they condition, to a large extent, the behavior of each profile.

### 3.1. Communication in the Care Encounter

The results of our study indicate that, in the healthcare encounter, the construction of an effective communicative space between the people involved in it is essential.

‘… *Communication is the main tool of my work; for me, teaching to care, not only to care but to motivate them to care, to pamper, to reinforce, to give a little support… and with the language barrier everything is still very technical*’.(I HP nº29 PC).

#### 3.1.1. Language Difficulty

The professionals we interviewed propose a reductionist approach to communicative conflict focused on the language barrier; in their own words, ‘*… I think the main barrier is not cultural, it’s language. The language wall*’. (I HP nº29 PC).

Women also consider that knowledge of Spanish is essential to be able to function in society. They live with a certain anxiety of not being able to express themselves and not understanding what health professionals tell them, ‘*… I have a lot of problems with the language. He who has no language has nothing here (…) When I go to the doctor I get very nervous, because I don’t know how to tell him anything*’. (I W nº11).

#### 3.1.2. Tools to Overcome Language Barriers

The Health Service of Castilla-La Mancha (SESCAM) has a telephone translation service and an automatic translation program that do not solve the difficulties that arise in care. There are professionals who say they have heard of the telephone translation service but have never used it. Those who know it and have used it on occasion comment that it does not solve their difficulties, ‘*… during childbirth you can’t have a phone hooked up there ‘look, count it’ because I can’t hold the phone in my hand*’. (I HP nº2 HC).

On many occasions, the information is given through written pamphlets, which are not always translated into Arabic and, furthermore, many of the women do not know how to read, so they depend on someone to translate them for them, ‘*… what happens is that there are many who can’t even read Arabic, so even if you give it to them, either their husband reads it or they don’t even know how to read it*’. (I HP nº15 HC).

We can interpret that the production of brochures translated into Arabic is an adaptation to the needs of people who do not speak Spanish, specific attention to specific needs. However, it only considers it from the point of view of one of the parties, since the reality of women is not taken into account, for whom the orality of communication is important, ‘*…we rely a lot on word of mouth*’. (I W nº45).

Professionals comment that they prefer that women go to consultations accompanied by people who know Spanish and translate for them. In most cases, it is the family who act as the interpreter. If it is children, we may encounter problems associated with gender roles that negatively condition and interfere with communication.

‘… *If they are daughters it is easier because you can pass on any information to them, but if they are sons, there are certain topics that you cannot touch because it is not right for them to translate things about bleeding… Then you run into a very serious problem’*.(I HP nº16 HC).

Another strategy used is to use other people, other than the patient, to help them with the translation, ‘*… you go out in the hallway and you say ‘you, do you know Spanish? So come here*’. (I HP nº43 PC).

This practice has become so normalized that Moroccan women themselves follow it, both when they are the ones who need translation and, when they are the translators, going so far as to say that not knowing the language is their problem,

‘… *when she can’t find anyone, she starts looking for a Moroccan woman in the hospital who can translate for her. Go out and look for someone who speaks Spanish. Because that’s your problem, if you don’t know something, go out in the street, look for someone and come back to the clinic*’.(FG2).

### 3.2. Interpersonal Relationships in the Healthcare Encounter

#### The Silence in the Meeting

In the study, silence and the low participation of women appear recurrently in the discourses of both profiles. However, the social representations underlying both behavioral patterns are not shared.

1.Perceptions and behaviors of health professionals

When interpreting silence, professionals identify several factors: the submission of women, ‘*… it’s all yes, they are very submissive, very respectful of what I can tell them, it’s not easy for them to ask me questions or tell me things*’. (I HP nº2 PC).

And the asymmetric relationship, their position of superiority, ‘*They don’t demand anything in particular, they go along with what we do to them (…) They don’t offer any alternative to what we tell them (…) because they are going to see us as very authoritative*’. (I HP nº9 HC).

They also refer to the support networks (family and friends) that act as guides and provide women with the information they need, ‘*… they are not very likely to ask questions… because they sort it out among themselves*’. (I HP nº14 HC).

This silent behavior of women worries them and can lead them to make certain cultural impositions, which, although based on scientific evidence, push aside the attempt to understand women’s behavior and safeguard the right to autonomy in their decisions.

Thus, in order to protect their health and that of their babies, they adopt a series of paternalistic practices, ‘*… I don’t inform them because they (the vaccines) are very expensive, there are four doses and each dose costs 80 euros. So it creates a feeling of guilt, that they can’t get it*’. (I HP nº19 PC).

And infantilizing towards women, using expressions such as, ‘*look pretty, that’s how it’s done*’. (I HP nº14 HC), in a tone as if they were addressing a girl.

On the other hand, professionals explain that the difficulty of communication leads them to provide care focused on carrying out the techniques and procedures,

‘*They don’t play the same game as other women with whom you can tell more things, more details, they can tell you their fears, you can reassure them? That’s not possible; (…) in short, the conversation is much more simplistic ‘everything is fine’, ‘everything is wrong’, ‘it hurts’, ‘it doesn’t hurt’, very simple things*’.(I HP nº9 HC).

2.The Experience of Women in the Care Process

Due to the language difficulty, women remain silent for fear that their words will be misinterpreted and lead to misunderstandings,

‘*There are many women who are very quiet… They don’t want to talk. Some don’t have perfect language, they are afraid that when they say one word, it means another… They have gone to talk to the doctor and maybe they want to say one thing and the doctor will think something else*’.(I W nº45).

This attitude is reinforced by their perception of discriminatory treatment by professionals. They recognize that this situation significantly conditions their behavior in meetings, avoiding openly expressing their doubts and practices for fear of the reactions that these may provoke in health professionals, ‘*… Many people omit or avoid saying things because they know they may be prejudiced. For example, I can’t say that I am pregnant and that I am doing Ramadan*’. (I W nº46).

However, we find discourses from women, a minority, which aim to define the relationship as satisfactory.

On the other hand, the majority of the women interviewed express a feeling of misinformation in healthcare encounters that causes them uncertainty and fear that translates into silence and little participation. with which the misinformation is even greater,

‘*No, they haven’t explained what happened to me or anything. They have to give you explanations. Of course, and there are also controls (in PC), you have to advise these women to go to the controls… but there are many who don’t know about it*’.(I W nº45).

Respect for health professionals is another element that contributes to women maintaining a position of submission, shyness, and silence in meetings.

## 4. Discussion

This study aims to describe the interaction processes that occur between health professionals and Moroccan migrant women in reproductive healthcare, focusing on the communicative aspects and the interpersonal relationship that occurs in the care encounter.

There is a broad consensus in identifying communication as the pillar of the therapeutic relationship that facilitates or hinders the precision of the diagnosis, therapeutic adherence, and the involvement of patients in their own care [18].

In our study, language difficulty is identified as the main communication barrier, despite the fact that we know that communication is not only conditioned by language but also by cultural values, beliefs, and behaviors and by other barriers that can interfere and harm interpersonal relationships in the healthcare encounter: semantic distortion, poorly expressed messages, and poor patient care when they are blocked by their health problem [19].

However, we have found the same results in other studies similar to ours, in which the language barrier is considered one of the main challenges for effective access to healthcare for migrants and the main cause of low satisfaction with the healthcare attention received [20].

Regarding the tools to overcome situations of lack of communication to which both professionals and patients have access, satisfaction regarding their effectiveness in improving communication is limited. Our study highlights that, although telephone translation services, automatic translation programs, or written information in Arabic are available in health facilities, they are little used and are considered insufficient to solve the communication difficulties that arise in reproductive healthcare. Frequently, users and professionals choose to use informal translators [21,22] who may be family members, close friends, or even strangers.

While it is true that this practice is common, it is not advisable. The management of information in the reproductive process is complex. The topics addressed are especially sensitive to issues such as religion and gender roles, and, as we see in our study, are treated with little rigor from the point of view of the confidentiality of the Clinical History. The type of information that the Clinical History (Obstetric History) contains in the process of our research is a type of information that, in accordance with Organic Law 15/1999, of December 13, on the Protection of Personal Data (LOPD), constitutes specially protected information (Art. 7.3 of Organic Law 15/1999, of 13 December, on the Protection of Personal Data (LOPD): “Personal data that refers to racial origin, health and sexual life may only be collected, processed and transferred when, for reasons of general interest, a law so provides or the affected party expressly consents.”), and its handling without the express consent of the women may entail a sanction for whoever does so. One of the reasons that may be behind this fact may be the lack of knowledge of the legal significance of this practice on the part of healthcare professionals.

On the other hand, numerous investigations warn that working with informal translators can be even more dangerous than completely dispensing with interpretation since erroneous translations of diagnoses, treatments, and care can occur, causing misunderstandings and loss of confidentiality of clinical information [23]. Most of the evidence points to the need to have trained mediators instead of using informal and trained translators to reduce the possibility of clinical errors at the time of translation and the prevention of conflicts that lead to problems [24,25], especially in care in the perinatal stage [26]. Sánchez [27] states that the use of professional interpreters avoids the following: poorly given explanations; situations of misunderstanding and erroneous follow-up of treatments; unnecessary repetition of consultations; the performance of unjustified tests; and analysis and diagnostic errors, fundamentally.

In the results obtained on interpersonal relationships in the healthcare encounter, the category “silence in the encounter” stands out, understood as the silence and the low participation of Moroccan women in perinatal consultations. Obviously, the interpretation of this silence made by the women themselves and by professionals is not coincidental.

The women participating in the study argue their silence and low participation as a consequence of the difficulties they have in communicating, which leads them not to express their fears or doubts to avoid misunderstandings or tense situations. These results are similar to those obtained in a study on prenatal care of Somali women in Norway [28].

We believe it is important to mention that a large portion of immigrant women are illiterate and that, even if they have some command of the language, they often cannot read the informative documents that are provided to them, even if they are written in their mother tongue. To this situation, we must add that, for them, the transmission of knowledge is carried out orally [29,30] and this would only be possible if there was the intervention of intercultural mediators, a figure that is not common to find in the healthcare domain—the context where we have carried out our research.

Another aspect that deters participation is women’s perception of discriminatory treatment by professionals. As elements that could be at the basis of this type of treatment, we have found research that suggests that these discriminatory attitudes that our women point out may reflect the inability of health professionals to face cultural diversity and their lack of sensitivity in relation to the health beliefs of immigrants. In a Swedish study, midwives recognized the difficulties uniquely experienced by Somali women, arguing that they lacked appropriate resources in their prenatal care encounters [31]. The need to incorporate cultural competence in the training of health professionals as a strategy to combat cultural bias is widely documented [32,33]. It should be noted that, without downplaying the cultural component that is part of the social representations of women in terms of what the relationship with health professionals should be like (submission, respect) and the magnification of these factors as causes of conflicts in the healthcare encounter, this highlights a reductionist approach to caring for cultural diversity in the healthcare field.

In this research, the view of the participating health professionals is not neutral; it is filtered by these biases that condition the gaze itself, what is seen, the interpretation of what is seen, and what is achieved (the practices that materialize in the meeting). In the words of Aizenberg, “The view of culture understood as a homogeneous and static repertoire that collectively defines the identity of people and their behaviors has permeated the views of healthcare providers” [34].

The imprint of the Biomedical Model [35] has a lot to do with this approach to healthcare. The protocolization and standardization of care collides with the individual needs of people that go beyond cultural differences in the way of understanding care for health problems. In our study, health professionals report providing “more simplistic” care, limiting themselves to performing technical tasks and leaving aside the comprehensiveness of care. But this is not the only option that one can aim towards; in fact, there are experiences in which the care encounter in the perinatal stage is configured as a space of active listening and openness to the other with the aim of achieving mutual understanding [36], incorporating a comprehensive approach to communication beyond a simple tool to inform and disseminate messages. The challenge is great, but it is possible by combining new information technologies (translation programs), training, and interdisciplinarity in the communicative space [7], in which we once again highlight the figure of the intercultural mediator.

### Limitations

One of the limitations we encounter is related to access to women and their language limitations.

Regarding the methodology, it should be noted that the study was carried out in a certain geographical context in Spain and the sample was chosen to be representative, but we cannot guarantee that the results of this study can be applied or extrapolated directly to all Moroccan immigrant women in Spain.

However, we remember that qualitative studies are not based on the statistical representation of the sample, and they are not intended to achieve the representation of a population based on their conclusions.

Another limitation is related to the way in which the information was collected and the interaction between people in the interviews and focus groups. The type of relationship established can condition the collection of information, either because the characteristics of the interviewer influence the participants or due to the effects of social desirability.

Furthermore, we cannot ignore that despite the strengths of Grounded Theory, the theory generated depends to a large extent on the researcher’s interpretation and theoretical sensitivity, which can introduce biases, the most obvious of which is that the presentation of the findings is guided not only by the theoretical approach but also by the emerging content axes in the discourse.

However, it should be noted that Grounded Theory remains a rigorous and useful method for qualitative research, provided that the researcher is aware of its scope and knows how to address these issues adequately. The emphasis on the inductive generation of theory from the data remains a key strength of this method.

## 5. Conclusions

This study presents the interaction between health professionals and Moroccan migrant women in reproductive healthcare. The findings highlight that there are communication and relationship problems that alter healthcare and the professional–patient relationship.

Although both health professionals and Moroccan women are aware of the need for good communication to ensure quality healthcare, the available translation means (tele-translation, translation programs, etc.) are not used and informal translators are used, with various risks that these entails. In this sense, the implementation of the figure of a mediator/translator specializing in intercultural health is a priority.

In addition to the incorporation of this figure, to address disagreements in intercultural encounters, training of health professionals in cultural competence is configured as one of the fundamental strategies in order to reduce the stereotypes and uncertainties that, in the majority of occasions, lead to a series of behaviors on the part of professionals that are interpreted as discriminatory by the women who are the subjects of our study.

We consider that our research topic is of great importance in obtaining greater knowledge about the circumstances that condition attention to the reproductive health of Moroccan women. The strategic approach must be comprehensive, implementing measures at three levels (health professionals, Moroccan women and administration and institutions), configuring a series of actions that must first undergo an analysis of their viability.

## Figures and Tables

**Table 1 healthcare-12-01577-t001:** Characteristics of Moroccan immigrant women (*n* = 30).

Variables		Women’s Age (in Years)			Total
		21 to 30	31 to 40	41 to 50	
Residence time (in years)	<2	2	1		3
	3–5	3	2		5
	>6	7	7	8	22
Area of residence	Rural	7	8	5	20
	Urban	5	2	3	10
Education level	Primary	4	7	4	15
	Secondary	7	4	3	14
	University			1	1
Total		12	10	8	30

Source: Own elaboration by the authors.

**Table 2 healthcare-12-01577-t002:** Characteristics of health professionals (*n* = 35).

	Primary Care	Hospital Care	Total
Primary Care Nursing	2	-	2
Maternity Nursing	-	7	7
Pediatric Nursing	3	-	3
Midwives	4	10	14
Gynecology/Obstetrics	-	5	5
Pediatrics	4	-	4
Total	13	22	35

Source: Own elaboration by the authors.

**Table 3 healthcare-12-01577-t003:** Outline of content and script of questions for interviews and focus groups.

Themes	Professional Questions	Questions Women
Personal information	Sex Training Scope of work: AP/AE Service/Unit of work	Sex Studies Residence area Number of children Residence time in Spain
Interaction in the therapeutic encounter	What does it mean for you to work with these women? Do you feel comfortable with them? Do you think they understand you? Do you think you have the same goals? If you had to say, what is the degree of correlation between the needs that you perceive women have and what they really have? Do you think women notice that there is harmony between these perceptions (on your part and theirs)? How do you manage the difference between what you expect of them and what they do? (in the event that they do not coincide, of course) Why do you think this disagreement occurs? (language, culture,…)	Do you consider that the time they dedicate to you in each consultation is adequate? Why? Do you consider that the space where they serve you is adequate? If not, what would you change? Does the care they give you meet the expectations you have? If not: what are you missing (from each professional profile)? Can you say what you think, believe, need?
Communication	Do you have communication problems? Do you understand what they say to you? Do you think they understand what you say? Can you say what you think, believe, need? What resources do you know within the health system that facilitate communication? Is it easy to access these resources? Are they useful to you? Which one do you think is best? At what point in the process do you need them? What other resources could be implemented in the health system?	Do you have communication problems? Do you understand what they say to you? Do you think they understand what you say? If you have problems, what does the professional do? What resources do you know within the health system that facilitate communication? Is it easy to access these resources? Are they useful to you? Which one do you think is best? At what point in the process do you need them? What other resources could be implemented in the health system?

Source: Own elaboration by the authors.

**Table 4 healthcare-12-01577-t004:** Participant identification procedure.

▪ The following procedure has been followed to identify the speakers of each section of speech incorporated as an example. Each verbatim is followed by a code that is made up of four elements: ▪ The first is an alphabetical code that identifies whether it is an individual interview, a group interview, or a focus groups. ◦ I: individual interview ◦ GI: group interview ◦ FC: focus groups The second identifies the participant’s profile: ◦ W: woman ◦ HP: health professional ▪ The third is a number that numerically identifies each participant ▪ The fourth is a two-letter code to identify the field in which the participating health professional works: ◦ PC: Primary Care ◦ HC: Hospital Care

Source: Own elaboration by the authors.

**Table 5 healthcare-12-01577-t005:** Themes and categories identified after thematic analysis.

Topics	1	2
Interaction processes between health professionals and women in the perinatal period	Communication in the healthcare encounter	Interpersonal relationships in the healthcare encounter
Categories Emerging	Language Difficulty Tools to overcome language barriers	The silence in the meeting - Perceptions and behaviors of health professionals - The experience of women in the care process

Source: Own elaboration by the authors.

## Data Availability

The data are held in confidence by the lead author of this manuscript. They can be made available on request.

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
