# Peer review of "Interaction Processes between Health Professionals and Moroccan Immigrant Women in Reproductive Healthcare: The Disagreement in the Encounter—A Qualitative Study"

_healthcare, 2024, doi:10.3390/healthcare12161577_

Round 1
Reviewer 1 Report
Comments and Suggestions for Authors
Thank you for the opportunity to review the manuscript entitled: Interaction processes between health professionals and Moroccan immigrant women in reproductive health care: A Qualitative Study.
This is an important topic. Howver the presentation of the study in this manusckrtip makes it difficult to follow. There is significant information missing from the methods section which means that I could not assess the findings section or Disucssion and Conclusion. I suggest major revisions before being considered for publication. This includes editing the paper to improve readability and correct the tense used throughout.
I wish the authors all the very best with their revisions.
Abstract
Please revise the first paragraph and state the problem clearly such as,
Spain is a multicultural society. Moroccan women of childbearing age make up % of new migrants to the country.
Methods
Do you mean Descriptive qualitative study?
Introduction
Please revise the first paragraph. This should clearly frame the phenomenon under investigation by addressing 1. What the problem is, 2, who it impacts, 3 where and why it is important.
Line 47. Page 2. And please remove subjective language such as ‘in fact’
Lines 40 and 44. Please use the correct citation source and remove the intext links – unless this is what the journal has requested.
Citation is needed at the end of the sentence line 101-102 at the end of word ‘dimensions’.
Line 106, p, 3. Do you mean your aim was to explore the interaction process?
Methods
Please see my comment regarding how you have phrased on line 109, p 3. Method in the Abstract. I suggest you use the same terms as you have here in the Methods section for consistency.
It’s a little confusing here as you state multiple aims. Do you mean objectives- outcomes? Please revise.
Line 115- please start the sentence with text, not numbers. I suggest We sought a sample of…
And include inclusion and exclusion criteria please.
Please include How were participants selected? e.g. purposive, convenience, consecutive, snowball. How were participants approached? e.g. face-to-face, telephone, mail, email. Then state how many you recruited to the study.
How many refused? and reasons if possible.
Setting
Where was the data collected? e.g. home, clinic, workplace.
Tables 1 and 2 belong in the Results section. Please edit.
Please provide a description of the Researchers’ characteristics that may influence the research, including personal attributes, qualifications/experience, relationship with participants, assumptions, and/or presuppositions; potential or actual interaction between researchers’ characteristics and the research questions, approach, methods, results, and/or transferability.
Data collection
Please describe how the data were collected clearly; include the details of data collection procedures including (as appropriate) start and stop dates of data collection and analysis, iterative process, triangulation of sources/methods, and modification of procedures in response to evolving study findings. This may include a pilot of the interview questions. Was this done and if so were any changes made to the questions and types of data collected and devices (e.g., audio recorders) used for data collection?
Line 138. I am not sure what you mean by the Information saturation criterion” More information is needed here including a citation.
Line 138. Page 4. When you refer to ‘discussion groups’ do you mean group interviews or focus groups?
Data processing and analysis
This section is not clear. The reader should be able to read this and follow your process and repeat it if necessary.
Please include in this order the,
· Methods for processing data prior to and during analysis, including transcription, data entry, data management and security, verification of data integrity, data coding, and anonymization/deidentification of excerpts.
· The process by which inferences, themes, etc., were identified and developed, including the researchers involved in data analysis; usually references a specific paradigm or approach.
· Techniques to enhance the trustworthiness and credibility of data analysis (e.g., member checking, audit trail, triangulation).
.
Line 153 p 5. Citation is needed for Atlas-Ti, please
Line 158. Please clarify who ‘everyone’ refers to and as suggested earlier you need to include the credentials of each member involved in the research
Line 169-The section starting with “Before …. Needs to be moved to data collection please.
How did you guarantee anonymity?
Result
Please summarise the participants' characteristics and state the sample. It seems some demographic data was collected. Please include how and why this was done in the methods under data collection.
Please summarise the main findings at the start of this section along with the characteristics as per above.
This section requires editing and rewriting, please. The overuse of quotes makes it difficult for the reader to understand the themes and how they emerged from your analysis. I suggest using indirect and direct quotes. The reader should be able to clearly understand the main themes and see how you go there. Quotes etc are then used as evidence to support your description and interpretation of the meanings.
Cirec quotes under 40 words belong within the sentence and only those >40 words are indented and separated from the paragraph.
Avoid using double quotes such as lines 228-235. Avoid finishing the theme with a quote. this leaves the reader to interpret your findings, which is the research team's role.
Discussion
A good summary to start the Discussion stating the main findings. Though this section hard to assess due to the lack of information in the Methods section.
Limitations- qualitative research does not seek to be representative of a population. please consider revising this section to include limitations relevant to the methods used – grounded theory.
Comments on the Quality of English Languagepoor grammar which detracts from the readability and content
Author Response
REVIEWER COMMENTS 1
Thank you for the opportunity to review the manuscript titled: Interaction processes between health professionals and Moroccan immigrant women in reproductive health care: a qualitative study.
This is an important issue. However, the presentation of the study in this manuscript makes it difficult to follow. There is significant information missing from the methods section, meaning I was unable to evaluate the findings section or the Decision and Conclusion. I suggest major revisions before being considered for publication. This includes editing the document to improve readability and correct the tense used throughout.
I wish the authors all the best with their reviews.
RESPONSE TO REVIEWER COMMENTS
We thank you for the work you have done reviewing our manuscript. Your comments have helped us improve our proposal, we have learned from your experience and knowledge that you have generously transferred to us.
We have incorporated our comments in green, both in this document and in the manuscript to facilitate their location and reading. Below, we respond to each of your opinions. We hope they meet your expectations and we remain attentive to what you want to tell us if you consider it appropriate.
Abstract
Please review the first paragraph and state the problem clearly, e.g.
Spain is a multicultural society. Moroccan women of childbearing age constitute % of new migrants to the country.
We appreciate the reviewer's suggestion and incorporate new wording (lines 16-18).
Methods
Are you referring to the qualitative descriptive study?
We have modified the wording of this section (line 20).
Introduction
Please review the first paragraph. This should clearly frame the phenomenon being investigated by addressing 1. What the problem is, 2, who it affects, 3 where and why it is important.
Line 47. Page 2. And please remove subjective language like "in fact"
We have modified the wording of this section (lines 41-43).
Lines 40 and 44. Please use the correct citation source and remove in-text links, unless this is what the journal has requested.
We have modified the wording of this section (lines 42-43).
The citation is needed at the end of sentence line 101-102 at the end of the word ' dimensions'.
We have incorporated quote (13)- (line 99).
Line 106, p, 3. Do you mean that your goal was to explore the interaction process ?
Yes, it is appropriate and consistent with this type of methodological approach, since it reflects an open, flexible approach oriented toward a deep understanding of a phenomenon.
Orientation towards understanding: The objective aims to understand the interaction process, rather than measuring or predicting it.
Methods
Please see my comment on how you worded it on line 109, page 3. Method in abstract. I suggest you use the same terms you have here in the Methods section for consistency.
We have modified the wording (lines 106-108).
It's a little confusing here, as it sets up multiple objectives. Do you mean objectives-results? Please review.
We have modified the wording (lines 110-111).
Line 115: Start the sentence with text, not numbers. I suggest we look for a sample of...
We have modified the wording (line 115).
And include inclusion and exclusion criteria, please:
We have incorporated the inclusion and exclusion criteria (lines 119-135)
Please include how participants were selected? for example, intentional, convenience, consecutive, snowball How did you approach participants? for example, face to face, telephone, mail, email.
We have incorporated this information (lines 141-143)
Next, indicate how many people you recruited for the study.
We have already provided this information.
How many refused? and reasons, if possible.
We do not know if any of the people contacted refused to participate in the recruitment process.
Adjustment
Where was the data collected? for example, home, clinic, workplace.
We have incorporated this information (lines 144-146)
Tables 1 and 2 belong to the Results section. Please edit.
With all due respect to your considerations, the argument that supports our decision to leave these tables in the methodology section is that in qualitative studies, detailed information about the participants is essential to establish the transparency and rigor of the research process. . The participant table makes it easier for the reader to evaluate the adequacy of the sample and the representativeness of the data collected, in addition to making explicit the criteria used to select participants, which is essential in grounded theory methodology and other approaches. qualitative based on theoretical or rich sampling .
Provide a description of the researchers' characteristics that may influence the research, including personal attributes, qualifications/experience, relationships with participants, assumptions and/or presuppositions; potential or actual interaction between researcher characteristics and research questions, approach, methods, results and/or transferability.
We have incorporated this information (lines 147-149)
Data Collect
We have incorporated this information (lines 151-154)
Clearly describe how the data was collected; include details of data collection procedures, including (as applicable) start and end dates of data collection and analysis, iterative process, triangulation of sources/methods, and modification of procedures in response to the evolution of the study findings. This may include pilot testing interview questions. Was this done and, if so, were changes made to the questions and types of data collected and the devices (e.g., audio recorders) used for data collection?
We have incorporated this information (lines 158-167)
Line 138. Not sure what you mean by information saturation criterion" More information is needed here, including a citation.
The number of interviews and focus groups (45) was established using the saturation criterion -Say here that it is already explained and referenced. Using different data collection techniques (interviews and focus groups) and contrasting the information obtained contributes to achieving more robust saturation.
Line 138. Page 4. When you refer to “discussion groups,” do you mean group interviews or focus groups?
When we say Discussion Groups, we mean Focus Groups. However, we have replaced the term Discussion Group with Focus Group throughout the manuscript to make it more understandable in the English language.
- Data processing and analysis
This section is not clear. The reader should be able to read this and follow your process and repeat it if necessary.
Please include in this order the,
Methods for data processing before and during analysis, including transcription, data entry, data management and security, verification of data integrity, data coding, and anonymization / de-identification of extracts . Incorporated in the text- lines .
We have incorporated this information (lines 183-187)
The process by which inferences, themes, etc. were identified and developed, including the researchers involved in the analysis of the data; It usually refers to a specific paradigm or approach.
We have incorporated this information (lines 197-203)
Techniques to improve the reliability and credibility of data analysis (e.g., member checking, audit trail, triangulation).
We have incorporated this information (lines 204-209)
Line 153, page. 5. Appointment needed for Atlas-Ti please
ATLAS.ti 8 is a software tool designed for qualitative data analysis, especially useful for researchers and academics working with large corpora of text, audio, image or video data.
Line 158. Please clarify who "everyone" refers to and, as suggested above, you should include the credentials of each member involved in the investigation
We have incorporated this information (lines 204-205)
Line 169-The section starting with "Before... Needs to be moved to data collection please.
We have removed this paragraph from here and moved it to lines 161-165
How did you guarantee anonymity?
We have already explained it in section 2.3.
Result
Please summarize the characteristics of the participants and indicate the sample. It appears some demographic data was collected. Please include how and why this was done in the data collection methods.
As we have commented previously, in qualitative research that uses the convenience method for sample selection, this information is incorporated in the Methodology section.
Please summarize the main conclusions at the beginning of this section together with the characteristics indicated above.
We have incorporated this information (lines 230-236)
This section requires editing and rewriting, please. Excessive use of quotes makes it difficult for the reader to understand the themes and how they emerged from your analysis. I suggest using direct and indirect quotes. The reader should be able to clearly understand the main themes and see how it goes there. Quotes etc. are used as evidence to support your description and interpretation of the meanings.
Cirec quotes less than 40 words belong in the sentence and only those >40 words are indented and separated from the paragraph.
We have incorporated their indications to the extent possible to show the reader the speeches of the people participating in the study, since they legitimize the analysis of the results. We wanted to convey the voice of the participants.
Avoid using double quotes, such as lines 228-235. Avoid ending the topic with a quote. This leaves it up to the reader to interpret their findings, which is the role of the research team.
We have followed your instructions.
Discussion
A good summary to start the discussion with the main findings. Although this section is difficult to evaluate due to the lack of information in the Methods section.
We hope that with the additions we have made following your suggestions for improvement, the content of this section is adequate. The number of words allowed for the manuscript limits us in this sense.
Limitations: Qualitative research does not seek to be representative of a population. Please consider revising this section to include limitations relevant to the methods used: grounded theory. Incorporated in lines:
We have incorporated this information (lines 454-462)
Reviewer 2 Report
Comments and Suggestions for Authors
Overall an interesting paper. Please see below for feedback:
- I noticed at least two URLs in the body of your paper. These should be in your references, not in the main text.
- Where do the majority of the immigrants to Spain originate?
- Why did you focus on Morocco? (Readers could hazard a guess, since Morocco is geographically very close to Spain, but please state the reason anyway so you don't need to make your readers guess.)
- Page 3, line 115: when you state "30 immigrant women from Morocco participated", please write out the number 30 as thirty, since it is at the beginning of the sentence.
- How did you recruit your study participants? Where did you recruit them? Are they from across Spain or do they live in a specific region?
- How often did you use a translator for your interviews?
- It would be easier to read the quotations if there was some space between them and the rest of the text.
- Also, subsections in the results are difficult to identify because they occur immediately after the preceding sections.
- You mentioned that data collection occurred between April 2020 and December 2023. How did COVID-19 impact your work? For example, were the Moroccan women able to bring family members with them as translators early in the pandemic? How did that impact the language barrier?
- Were the women covered by the Sistema Nacional de Salud?
- Did the women face any discrimination due to their religion?
Author Response
REVIEWER COMMENTS 2
Overall, an interesting article. See below for comments:
RESPONSE TO REVIEWER COMMENTS
We thank you for the work you have done reviewing our manuscript. Your comments have helped us improve our proposal, we have learned from your experience and knowledge that you have generously transferred to us.
We have incorporated our comments in green, both in this document and in the manuscript to facilitate their location and reading. Below, we respond to each of your opinions. We hope they meet your expectations and we remain attentive to what you want to tell us if you consider it appropriate.
- I noticed at least two URLs in the body of your article. These should be in your references, not in the main text.
We have corrected this error in the manuscript (lines 41-43)
- Where do the majority of immigrants in Spain come from?
- Why did you focus on Morocco? (Readers might hazard a guess, since Morocco is geographically very close to Spain, but state the reason anyway so you don't have to keep your readers guessing.)
The choice of this group of immigrant population was determined by the sociodemographic characteristics of the area where the study is carried out and by the data reflected on the portal of the National Institute of Statistics, in addition to the interest of the research group in this profile.
- Page 3, line 115: When you write down "30 immigrant women from Morocco participated," write the number 30 as thirty, since it is at the beginning of the sentence.
We have corrected this error in the manuscript (line 115)
-How did you recruit your study participants? Where did you recruit them? Are they from all over Spain or do they live in a specific region?
This information is incorporated into the manuscript (lines 119-135)
- How often did you use a translator for your interviews?
We had the help of two translators, although most of the women interviewed had a good command of the Spanish language, so their intervention was not much.
- It would be easier to read the quotes if there was some space between them and the rest of the text.
- Additionally, subsections of the results are difficult to identify because they occur immediately after the previous sections.
We have resolved this issue, thank you.
- You mentioned that data collection occurred between April 2020 and December 2023. How did COVID-19 affect your work? For example, were Moroccan women able to bring in family members as translators at the beginning of the pandemic? How did that affect the language barrier?
It was an error in the writing in the month in which the interviews began, since they began to be carried out in the month of June 2020. At that time, the security measures due to COVID-19 allowed proximity while safeguarding the recommended health measures. .
- Were women covered by the National Health System?
Yes, the current health regulations in Spain guarantee care for all immigrants residing in Spain.
- Did women suffer any type of discrimination due to their religion?
None of the women participating in the study stated that they had suffered discrimination for religious reasons.
Round 2
Reviewer 2 Report
Comments and Suggestions for Authors
I'm happy with the revised version of this manuscript.